# Research on Large-Scale Urban Shrinkage and Expansion in the Yellow River Affected Area Using Night Light Data

**Wenhui Niu** [1,†]**, Haoming Xia** [1,2,†]**, Ruimeng Wang** [1]**, Li Pan** [1]**, Qingmin Meng** [3]**, Yaochen Qin** [1,2]**, Rumeng Li** [1]**, Xiaoyang Zhao** [1]**, Xiqing Bian** [1] **and Wei Zhao** [1,2,*]

1   Ministry of Education Key Laboratory of Geospatial Technology for the Middle and Lower Yellow River Regions, Henan Key Laboratory of Earth System Observation and Modeling, College of Environment and Planning, Henan University, Kaifeng 475001, China; Niuwh@henu.edu.cn (W.N.); xiahm@vip.henu.edu.cn (H.X.); wangrm@henu.edu.cn (R.W.); panli970611@henu.edu.cn (L.P.); qinyc@henu.edu.cn (Y.Q.); Lirm@henu.edu.cn (R.L.); zhaoxy@henu.edu.cn (X.Z.); 104754200172@henu.edu.cn (X.B.)
2   Key Research Institute of Yellow River Civilization and Sustainable Development and Collaborative Innovation Center on Yellow River Civilization jointly built by Henan Province and Ministry of Education, Henan University, Kaifeng 475001, China
3   Department of Geosciences, Mississippi State University, Starkville, MS 39762, USA; QMeng@geosci.msstate.edu
*   Correspondence: 10130056@vip.henu.edu.cn; Tel.: +86-371-2388-1858
†   These authors contributed equally to this work.

**Abstract:** As the land use issue, caused by urban shrinkage in China, is becoming more and more prominent, research on urban shrinkage and expansion has become particularly challenging and urgent. Based on the points of interest (POI) data, this paper redefines the scope, quantity, and area of natural cities by using threshold methods, which accurately identify the shrinkage and expansion of cities in the Yellow River affected area using night light data in 2013 and 2018. The results show that: (1) there are 3130 natural cities (48,118.75 km$^2$) in the Yellow River affected area, including 604 shrinking cities (8407.50 km$^2$) and 2165 expanding cities (32,972.75 km$^2$). (2) The spatial distributions of shrinking and expanding cities are quite different. The shrinking cities are mainly located in the upper Yellow River affected area, except for the administrative cities of Lanzhou and Yinchuan; the expanding cities are mainly distributed in the middle and lower Yellow River affected area, and the administrative cities of Lanzhou and Yinchuan. (3) Shrinking and expanding cities are typically smaller cities. The research results provide a quick data supported approach for regional urban planning and land use management, for when regional and central governments formulate the outlines of urban development monitoring and regional planning.

**Keywords:** night light data; urban shrinkage; urban expansion; natural city; Yellow River affected area

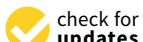

## 1. Introduction

Shrinkage is a phenomenon ubiquitous in natural and social systems. Many scholars have conducted research in the context of biology, economics, demography, and even computer science and social networks [1–8]. "Shrinking City" is the direct translation of the German term "schrumpfende Stadt", which is used to describe the phenomenon of population reduction and economic recession of German cities during the process of deindustrialization [9]. The phenomenon of real estate vacancy and infrastructure waste is the typical characteristic of urban shrinkage [10–12]. On the other hand, the research on expanding cities has never stopped, but much less attention has been paid to urban shrinkage. Since 1978, China has experienced a rapid and largescale urban expansion process [13,14]. This rapid urban expansion has led to some environmental and ecological

problems [15–17], such as the impact on urban vegetation productivity, urban water evolution, urban heat island effect, etc. These problems have an impact on the modernization process of Chinese cities and towns, and have attracted the attention of relevant experts and researchers [18,19]. In recent years, with the advances of big data technology, more scholars use remote sensing big data to study urban built-up areas, vegetation, water, and other aspects [20–22]. In general, previous studies on urban shrinkage and expansion mainly rely on population data and statistical yearbooks [23,24]. However, these traditional data have some limitations, such as difficulties in collection, inaccurate data statistics, different statistical standards, long cycles of updates, and rough spatial expression [25,26], which lead to the deviation of identified urban shrinkage and expansion. Facing the problems of the data, and the technical processes of identifying shrinking and expanding cities, some scholars have recently begun to use the night light synthesis data obtained by satellites to identify shrinking and expanding cities [27–29]. The artificial light observed by satellites at night provides us with effective alternative measurement for various human activities from local, to regional and global scales in the past few years [25,30–37]. Compared with other remote sensing observation methods of visible light, near-infrared or radar sensors [38–43], artificial night lighting data provides a unique perspective of lighting intensity, closely related to socioeconomic activities and urban development dynamics [44]. Therefore, night light data can be used to detect the spatial dynamics of different populations and socioeconomic activities around the world [45–49].

So far, using Suomi National Polar-orbiting Partnership (NPP) Visible Infrared Imaging Radiometer Suite (VIIRS) Day/Night Band (DNB) data, urban studies have focused on the following areas: (1) the relationship between NPP–VIIRS nighttime lights data (NTL) and socioeconomic indicators, such as population, GDP, and housing vacancy rate [50,51]; (2) the exploration of the spatial distribution of population and the spread of certain diseases [52]; and (3) the mapping of built-up areas and dynamically monitoring of urban expansion by using NPP–VIIRS NTL data [8,44,53–57]. At present, few studies have been focused on both urban shrinkage and expansion at the same time, and the related research needs to be strengthened. There are two main NTL data processing methods currently used for urban shrinkage and expansion research. One is to identify urban shrinkage and expansion by calculating the difference in the NTL radiation values of each grid in different years, but this difference method cannot determine the continuity and trend of urban shrinkage and expansion [27,28]. The other is to use NPP–VIIRS NTL data to calculate the changing slopes of the NTL radiation values of each grid to identify the shrinking and expanding of a city [58]. However, apparent noise may exist in the NTL data, and the study scope is concentrated on an entire prefecture-level city, rather than urban areas, and the study results may not be accurately representative of urban shrinkage and expansion. Therefore, our proposed study conducts simultaneous research on both urban shrinkage and expansion based on night light data.

The Yellow River affected area refers to the geographical area that is hydrologically affected by the Yellow River Basin, including the flooding areas and agricultural irrigation areas of the Yellow River [59]. The main purpose of this study is to identify the shrinkage and expansion of natural cities in the Yellow River affected area by calculating the variation rate of night lighting data. First, POI and road network data were used to define the boundaries of natural cities. Second, the changing rate of night light was calculated and validated with LandScan population data. Finally, the shrinkage and expansion of a natural city are identified according to the calculated results. We hope to provide a new perspective for the detailed and comprehensive identification of the shrinkage and expansion of natural cities by applying NTL data analytics to a big river basin, so as to provide a basis for urban development governance in the Yellow River affected area, which would provide regional urban development information for ecological protection and high-quality socioeconomic development in the Yellow River basin.

## 2. Materials and Methods

### *2.1. Study Area and Data*

#### 2.1.1. City System in China

Traditionally, cities are determined using administrative boundaries, such as provinces, municipalities, prefecture-level cities, and county-level cities in China. However, since the Chinese economic reform, urban systems are complicated and consist of strongly interconnected parts, including human networks and their connections with buildings and the natural environment [60–62], which often spread across administrative boundaries. In addition, intensified by current rapid global urbanization, urban morphology and agglomeration of functional areas have undergone tremendous changes in the past few decades [63,64], and especially in China, this change is more obvious [65–67]. Therefore, there are inconsistencies between the administrative boundaries of traditional cities and the real central cities.

In this study, a natural city is closely connected in space, with a complete built-up environment and infrastructure of the urban center area, which contains a minimum area of 2 km$^2$ [68]. therefore, a natural city covers the center of a city. Compared with the traditional research based on administrative boundaries, we can more accurately express the development status of cities through research on natural cities, and at the same time can distinguish and describe more clearly the unbalanced development of different areas within the same administrative unit, especially a large city area. Focusing on natural cities identification and mapping, both shrinkage and expansion characteristics of cities in the Yellow River affected area are studied using the satellite nighttime light data.

#### 2.1.2. Study Area

The Yellow River affected area covers 13 provinces and municipalities directly under the central government in eastern, central, and western China, including 531 cities and counties (Figure 1a). The total area of the Yellow River affected area is 1,412,900 km$^2$, of which the natural drainage area is 752,000 km$^2$. The region spans four geomorphic units (the Qinghai–Tibet Plateau, the Inner Mongolia Plateau, the Loess Plateau, and the Huang–Huai–Hai Plain) from west to east, and includes a variety of climate types and different levels of socioeconomic development. The region is one of the most densely populated areas in the world, and its human and socioeconomic activities are extremely intensive. On the one hand, with the implementation of the Western Development Strategy and the Central China Rise Strategy in the early 21st century, some cities have expanded rapidly. On the other hand, the consumption of coal and forests in the process of rapid industrialization has caused some cities that rely on these resources to deteriorate economically, resulting in urban shrinkage.

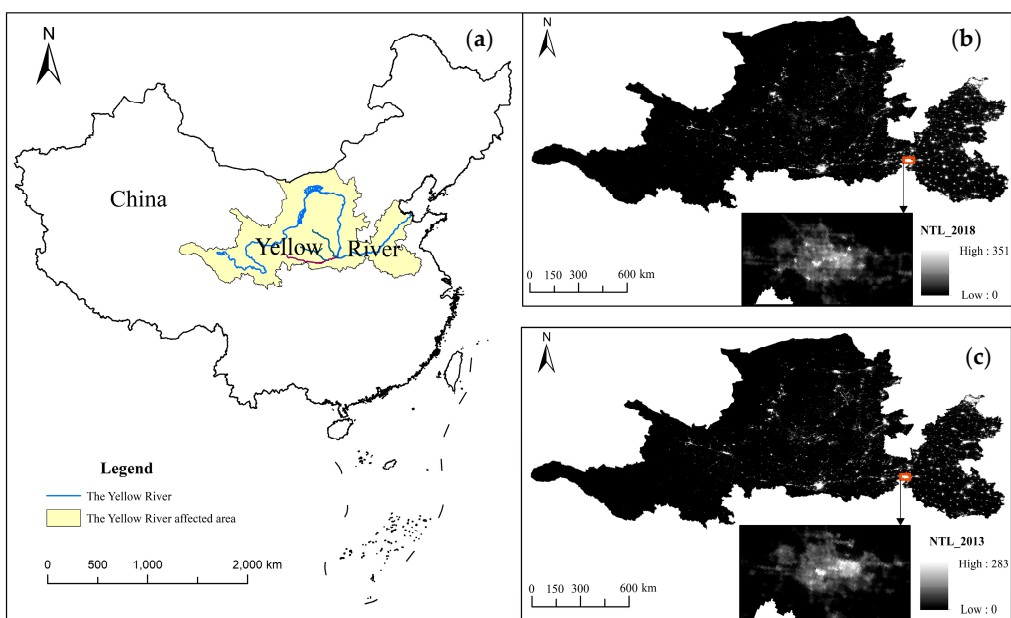

**Figure 1.** (**a**) The location of the Yellow River affected area and an overview of the study area; (**b**) Night light data of the Yellow River affected area in 2018; (**c**) Night light data of the Yellow River affected area in 2013.

### 2.1.3. Data Sources

This study uses five data types, including NPP–VIIRS data in 2013 and 2018 (http://www.ngdc.noaa.gov/eog/viirs/download_dnb_composites.html), LandScan data in 2013 and 2018 (https://landscan.ornl.gov/l.andscan-datasets), POI and road network data in 2018 (https://lbs.amap.com/), data types of POI and road network are shown in Table 1, and administrative boundaries of cities in 2018 (http://www.resdc.cn/Default.aspx).

**Table 1.** POI and road network data type example table.

| Name | Point of Interest | Road Network |
| --- | --- | --- |
| | Catering services | Highway |
| | Public facilities | National road |
| | Shopping services | Provincial road |
| | Road ancillary facilities | County road |
| | Companies and enterprises | Township road |
| | . . . | . . . |
| | Toponym and address information | Pedestrian road |

With the development of science and technology, multi-source data are used to study the development and changes in different fields. Multi-source big data are used to monitor the shrinkage and expansion of cities (Table 2). The night light images are NPP–VIIRS satellite data provided by the National Oceanic and Atmospheric Administration (NOAA). In fact, one of the earliest uses of Defense Meteorological Satellite Program (DMSP) data is to extract a city range [69], and the stable night light data of NPP–VIIRS can also effectively monitor the dynamic changes of a city [54]. Generally speaking, the demographic data can't accurately and effectively distinguish the floating population and the permanent population, and the time series of the statistical data are not continuous. Therefore, we use the LandScan data in 2013 and 2018 as the population data, which are provided by Oak Ridge National Laboratory (ORNL). These data are tailored to match the data conditions and geographical nature of each individual country and region (a community standard for global population distribution data), with an approximately 1 km spatial resolution.

**Table 2.** Basic information and main applications of urban multi-source data.

| Types of Data | Sources | Periods | Resolution | Application |
|---|---|---|---|---|
| Nightlight | NPP–VIIRS | 2013/2018 | 430 m | Identify the development model of the city |
| Population | LandScan | 2013/2018 | 1 km | Provide reference for night light data |
| Point of interests | Gaode LBS | 2018 | 50 points/km$^2$ | Defining the natural city |
| Road networks | Gaode API | 2018 | 600 m/km$^2$ | Modifying the urban boundary |

In 2018, there are 6,236,574 POI points and 2,851,714 road network data in the Yellow River affected area. Based on AutoNavi Map Location Based Services (LBS) and Application Programming Interface (API), the boundaries of natural urban areas and urban internal blocks are redefined to distinguish administrative cities.

2.1.4. Data Processing

In this study, the pseudo invariant feature method is used to identify the night light data to determine the detection threshold of the city [70]. Firstly, due to seasonal climate change and differences in human activities driven by weather, we selected the data of night light from January to April and July to October. Then the vector boundary of the Yellow River affected area is used to cut out the night light data of the 7 months every year. Finally, the annual average nighttime light intensity is obtained by averaging the data of 7 months after clipping (Figure 1b,c).

The VIIRS sensor of the NPP satellite improves the spatial resolution of the data. The resolution of sub-satellite point is 400 m, and there is no saturation phenomenon. However, the NPP–VIIRS imagery contains not only the permanent light intensity of cities, towns and other areas, but also the temporary light and background noise. Background noise could cause negative values in the imagery, and the surface of high albedo objects such as snow can make its pixel values in the imagery much higher than its true values, and, therefore, the imagery needs to be preprocessed.

Firstly, in order to avoid the impact of grid deformation and ensure that the area of the study region before and after the projection remains unchanged, referring to the WGS84 datum, this research converts the projection coordinate system of all images to Albers equal area projection and performs resampling. Secondly, the night light image is covered by the vector boundary of a natural city defined by POI data, and the night light image of the research area is then extracted. Previous studies have shown that there is a significant positive correlation between economy and lighting [71]. Beijing, Shanghai, Guangzhou, and Shenzhen are the most developed cities in China, which means that a pixel value in other regions cannot exceed the highest value of nighttime light in these four urban regions in theory [72]. If the pixel values in other areas are higher than the maximum nighttime light brightness value of these four megacities in China, they will be considered as abnormal values, which could be caused by the steady light of oil or gas fires [73]. Finally, the abnormal light brightness value caused by background noise is not filtered out in the light data [74,75], and therefore, a unified denoising process is needed. In this paper, we take $0.30 \times 10^{-9} \text{w} \cdot \text{cm}^{-2} \cdot \text{sr}^{-1}$ as the threshold to process the noise of night light data to obtain stable night light data [76,77]. The abnormal values in the data are processed as below in Equation (1):

$$DN_{(n,i)} \begin{cases} DN_{(n,i)} = DN_{(n,k)}, \ DN_{(n,i)} > thr_{max} \\ DN_{(n,i)} = DN_{(n,i)}, \ thr_{min} \leq DN_{(n,i)} \leq thr_{max} \\ DN_{(n,i)} = 0, \ DN_{(n,i)} < thr_{min} \end{cases} \quad (1)$$

In Equation (1), $DN_{(n,i)}$ represents the radiation value of the *i*-th pixel in the *n*-th year, and $DN_{(n,k)}$ represents the maximum radiation value of the eight pixels directly adjacent to the *i*-th pixel ($DN_{(n,k)} \leq thr_{max}$, if the adjacent pixel values are greater than $thr_{max}$, the maximum value of eight adjacent pixels in each pixel in the adjacent eight adjacent regions is selected). After this process, the pixel values of the corrected NPP–VIIRS data are all less than $thr_{max}$, and greater than zero.

In this study, we have quantitatively identified and described the shrinkage and expansion of cities in the Yellow River affected area in 2013 and 2018. Firstly, we use POI data and road network data to define a natural city. Secondly, on the basis of a natural city, the NPP–VIIRS data are processed by pseudo invariant features to identify the change of night light, and the results are compared with Landscan data. Finally, the shrinkage and expansion of a city can be defined by the change rate of night light (Figure 2).

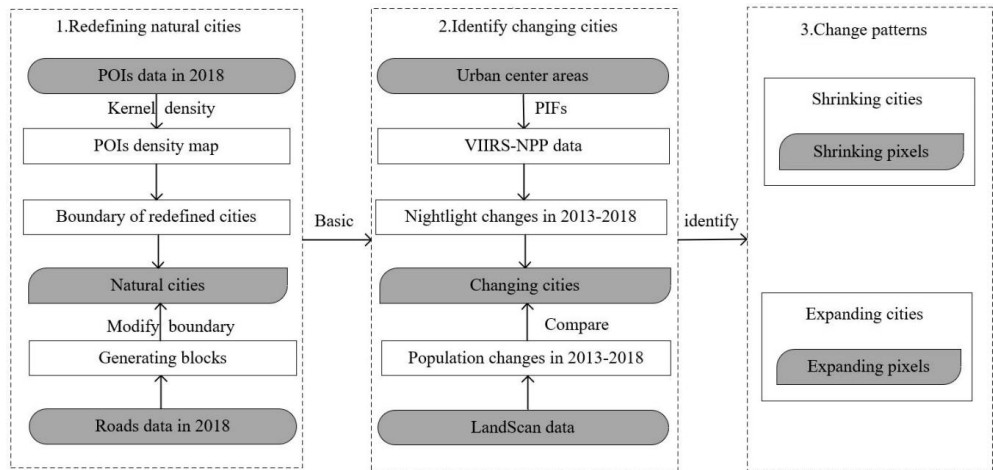

**Figure 2.** Analytical framework.

*2.2. Methods*

2.2.1. Redefining Natural Cities

Since the administrative boundaries of cities cannot accurately reflect the urban central areas, where human and socioeconomic activities are intensive, more scholars have recently used multi-source data fusion driven methods to study real urban systems [78–83]. In data driven research, urban central areas are identified and defined by multi–source data representing human activities and socioeconomic activities. Multi-source data includes remote sensing images [84,85], population density [86], built–up areas [87], road network [88], points of interest [86], and geotagged social media data [89]. In the above studies, urban boundaries are generally defined as concentrated areas of human activities and active urban centers [78]. In order to correctly characterize urban shrinkage and expansion, this paper redefines a city to reflect its real and natural central area. In other words, a natural city is a natural and objective description of urban scope according to the density of human settlements and activities [90,91].

The process of redefining a city is to use POI data to generate the boundaries of the urban center area of a natural city. Figure 3 shows the process of redefining the city of Zhengzhou as an example, including the following five steps. Firstly, the density of points of interest is calculated, and the density map of interest points with spatial resolution of 500 m is generated by using kernel density function in ArcGIS. Secondly, according to the study of Song, the optimal threshold value of POI density is 50 points/km$^2$, and the areas with POI density more than 50 points/km$^2$ are selected [68]. Then, the raster data is converted into vector data, according to Song's research results, the areas with an area greater than 2 km$^2$ are selected and defined as a natural city. Finally, the boundary of the natural city is modified with the road network data. The road data corrects some abnormal city boundaries and divides the interior of the city in more detail.

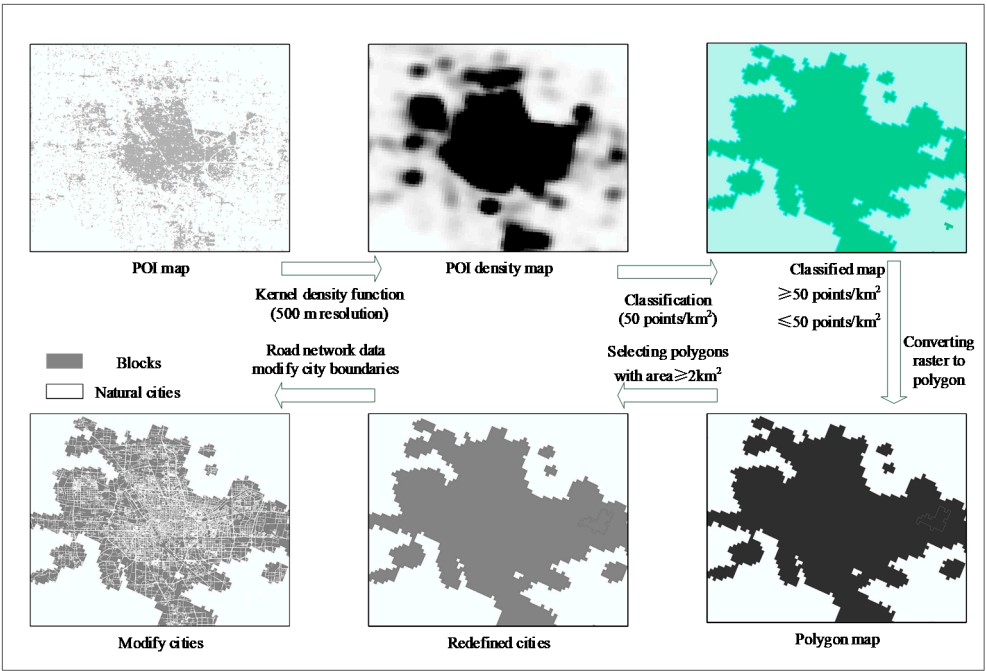

**Figure 3.** A diagram of POI-based method to redefine cities: an example of a natural city of Zhengzhou, a prefectural level city in Henan province, China.

2.2.2. Identify Shrinking and Expanding Cities

On the basis of redefining a natural city, the shrinking and expanding cities are identified by revising the NPP–VIIRS data in 2013 and 2018. The average values of raster data or vector data of each natural city are stored in a polygon, and the change ratios between the initial year and the end year of night light are calculated in the shrinking and expanding cities. The night light data of cities in the Yellow River affected area are masked and noise removed in ArcGIS, and then the light change rates of all cities are calculated. The calculation equation is as follows:

$$r_1 = \frac{\text{NTL}_{2018} - \text{NTL}_{2013}}{\text{NTL}_{2013}} \times 100\% \tag{2}$$

$$r_2 = \frac{\text{POP}_{2018} - \text{POP}_{2013}}{\text{POP}_{2013}} \times 100\% \tag{3}$$

In Equation (2), $r_1$ represents the rate of change of light, and NTL represents the brightness of the night light. In Equation (3), $r_2$ represents the rate of change of light, and POP represents population density.

We have calculated the slope of the radiance change of each pixel, and then define the shrinkage and expansion of a city based on the slope value. The measurement of 500 m × 500 m pixel scale can be classified into five categories: significant expansion ($r_1 \geq 0.30$), mild expansion ($0.10 < r_1 < 0.30$), stable ($-0.10 \leq r_1 \leq 0.10$), mild shrinkage ($-0.30 < r_1 < -0.10$), and severe shrinkage ($r_1 \leq -0.30$) [92]. With the same thresholds, we also used LandScan population data for 2013 and 2018 to compare the results different data sources and the potential connections between them.

## 3. Results

### 3.1. Redefined Cities Interpreted from POIs and Roads in 2018

Based on the POI and road network data in 2018, we finally redefined 3130 natural cities in the Yellow River affected area (Figure 4). These natural cities include prefecture-level and county-level administrative cities in China, covering an area of 48,118.75 km$^2$, accounting for 0.50124% of China's land area. Compared with the administrative cities

published by the state, the number of natural cities is obviously more, and there are a number of natural cities within an administrative city. The higher the administrative level of a city, the more natural cities are identified within it. The distribution of natural cities in the Yellow River affected area is uneven from east to west. The larger natural cities are mainly agglomerated in the central and eastern regions, while the natural cities in the western region are relatively small that is different from the traditional administrative urban areas.

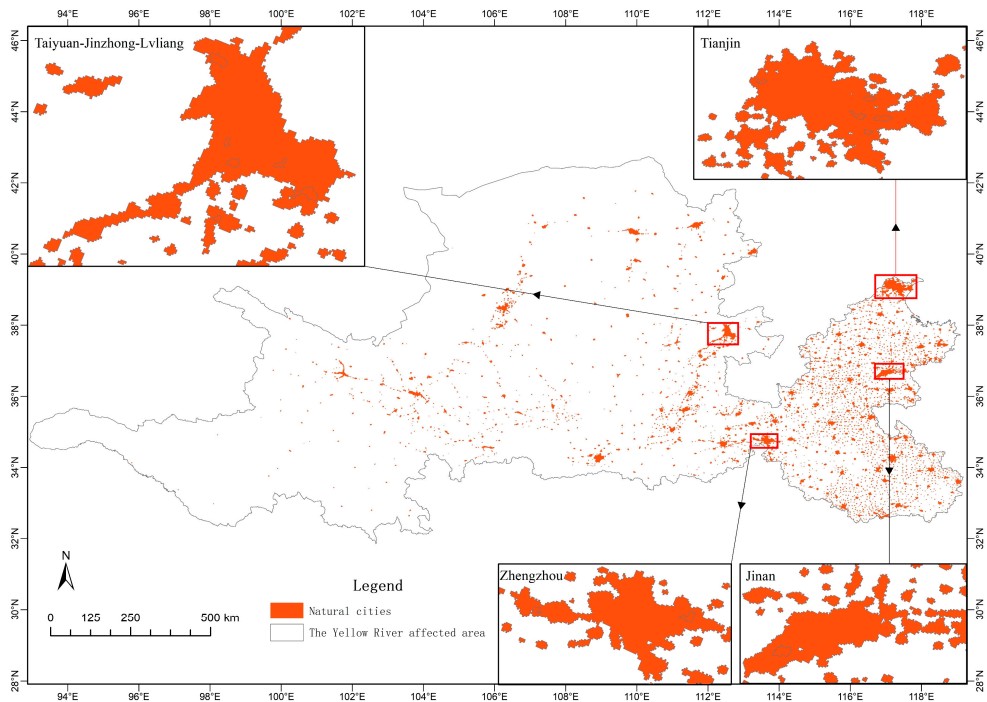

**Figure 4.** Natural cities distribution throughout the study area.

It is the first time to spatially analyze and map the development of natural cities in the Yellow River affected area. The largest natural city is located in the south of the administrative area of Tianjin (2103.25 km$^2$); the second largest natural city is located at the junction of Taiyuan, Jinzhong, and Lvliang (957.50 km$^2$); the third natural city is located in the administrative area of Zhengzhou (833.00 km$^2$); and the fourth natural city is located in the administrative area of Jinan (816.75 km$^2$).

According to Zipf's law [71,93], Figure 5 shows the relationship between the area and the number of natural cities, where the number of natural cities decreases as the area increases. There are 2335 natural cities with an area of 2–10 km$^2$ (74.60%), 743 natural cities (23.74%) within 10–100 km$^2$, 48 natural cities (1.53%) within 100–500 km$^2$, and 4 natural cities with an area of more than 500 km$^2$ (0.13%).

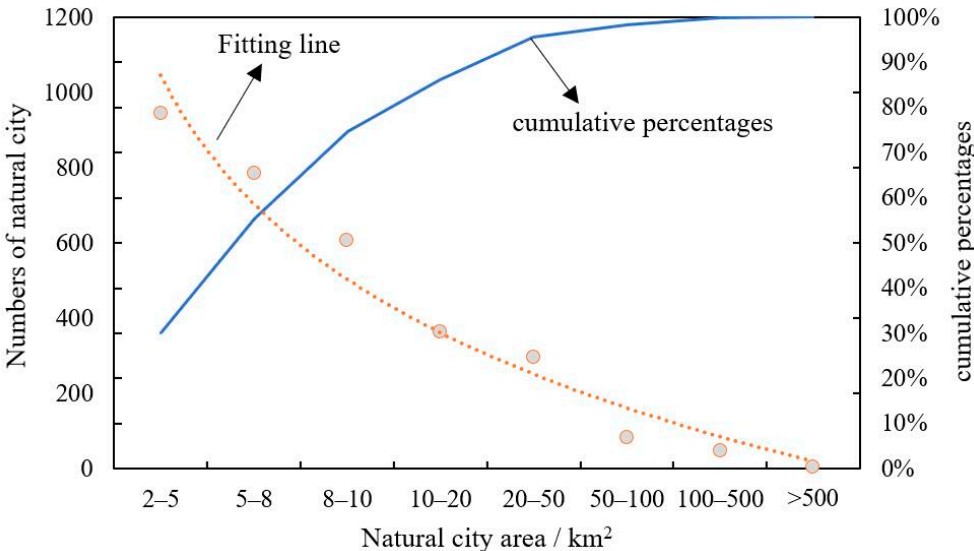

**Figure 5.** Relationship between natural city areas and the numbers of natural cities.

### 3.2. Spatial Distribution Pattern of Urban Shrinkage and Expansion in the Yellow River Affected Area

#### 3.2.1. Identified Shrinking and Expanding Cities Based on 2013–2018 NPP–VIIRS Data

Using NPP–VIIRS data in 2013 and 2018, 604 shrinking cities (19.30%) with a total area of 8407.5 km$^2$ were identified and mapped from 3130 natural cities; 2165 expanding cities (69.17%) with a total area of 32,972.75 km$^2$ were labelled and mapped; in addition, 361 stable cities (11.53%) with a total area of 6738.5 km$^2$ were identified. According to Equation (2), we calculate the change rate of cities. Figure 6a shows the spatial distribution of shrinkage and expansion of natural cities in the Yellow River affected area.

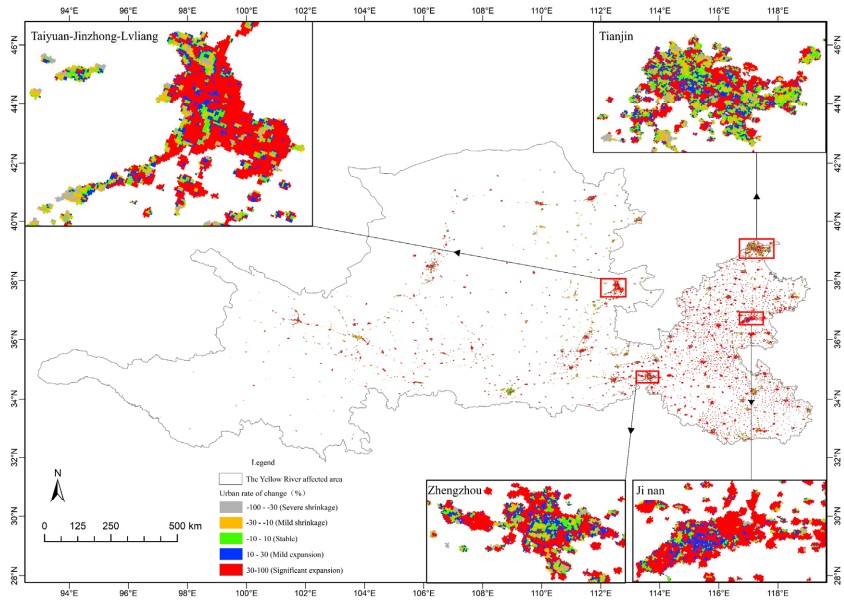

**Figure 6.** *Cont.*

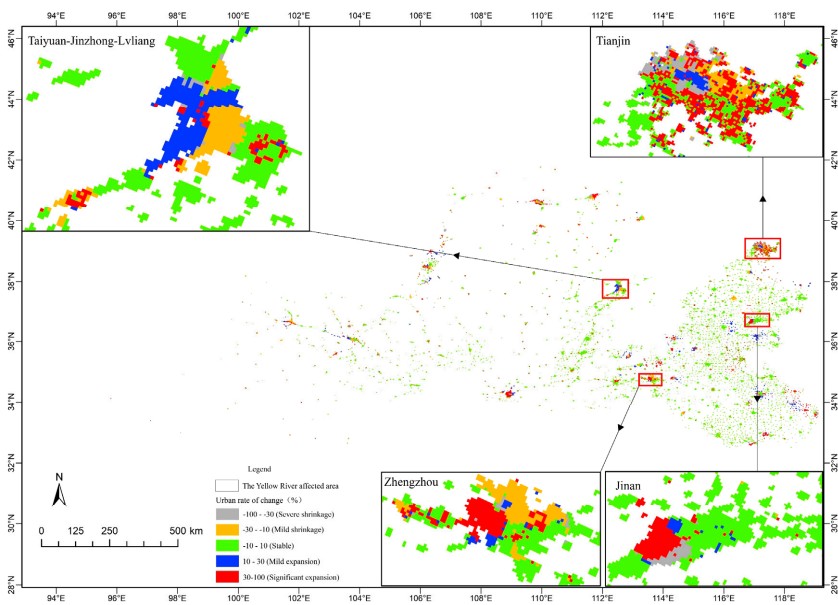

**Figure 6.** Spatial distribution of natural city shrinkage and expansion in the Yellow River affected area. (**a**) Shrinking and expanding cities with the shrinking and expanding ratio in nightlight data; and (**b**) Shrinking and expanding cities with the shrinking and expanding ratio in population data.

Meanwhile, compared with the results calculated by Landscan population data during the same period (Figure 6b), there is a difference between the number of shrinking and expanding cities and the results obtained by NPP-VIIRS data. The number of shrinking and expanding cities identified by NPP–VIIRS night light data are more than those identified by LandScan population data (Figure 7). Comparing the results of night light data and Landscan population data, we can see that the number of shrinking and expanding cities identified by population data is less than that by night light data, and the number of basically stable cities is more than that of night light data.

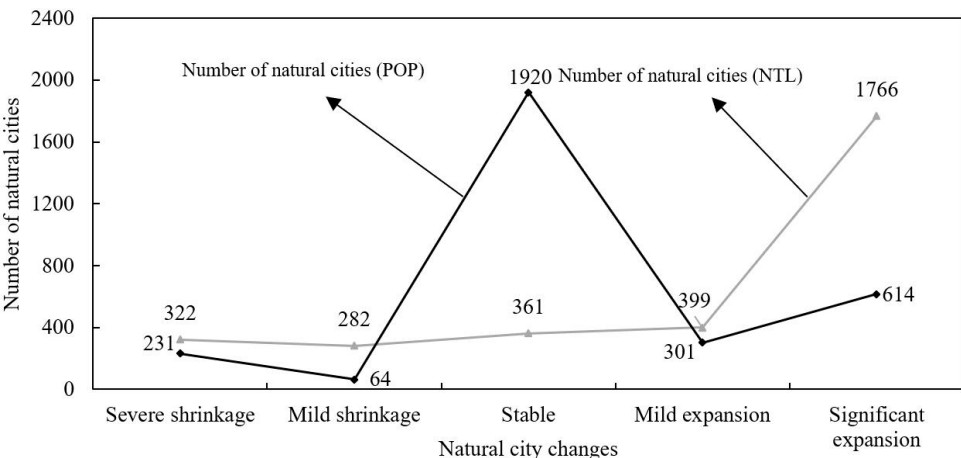

**Figure 7.** The relationship between urban shrinkage and expansion and the number of cities.

### 3.2.2. Overall Spatial Pattern of Shrinkage

The results of night light data analyses show that the shrinkage and expansion of natural cities are obvious in the Yellow River affected area from 2013 to 2018, and the urban development is unbalanced, with a combined contracting and expanding process. Figure 6 shows the spatial distribution of natural urban shrinkage and expansion in the Yellow River affected area. Shrinking cities are mainly distributed in the upper reaches of the Yellow River affected area except for the administrative cities of Lanzhou and Yinchuan,

and also in the middle and lower reaches of the Taiyuan–Jinzhong–Luliang city cluster, the surrounding areas of Tianjin Administrative Region and the junction of Shandong, Henan, and Anhui provinces. In addition, the areas around the administrative cities of Zhengzhou and Jinan have also experienced different degrees of shrinkage.

The shrinkage rate of natural cities in the Yellow River affected area reached 19.3% during the period from 2013 to 2018. Among the shrinking natural cities, the proportion of severely shrinking cities is 53.31%, of which only 2.48% are cities with an area of more than 100 km$^2$, and 97.52% are cities with an area of less than 100 km$^2$. In addition, the proportion of shrinking cities with an area of more than 10 km$^2$ is only 32.30%, and that with an area of less than 10 km$^2$ is 67.70%. Cities with mild shrinkage accounted for 46.69%, of which only 1.06% are cities with an area of more than 100 km$^2$, and 98.94% are cities with an area of less than 100 km$^2$. In addition, only 29.08% of the cities with an area of more than 10 km$^2$ and 70.92% of the cities with an area of less than 10 km$^2$ (Table 3). It can be seen that most of the shrinking cities are cities with smaller areas, which could be mainly due to the small urban area, backward resources in various aspects, difficulties in attracting investment, and talent shortages. In addition, as large cities could attract more capital investment and talent gathering, the economic vitality of cities will burst out, while the small cities around big cities have no core competitiveness, talent or capital flows into the surrounding large cities, which eventually lead to that small city gradually shrinking.

**Table 3.** Shrinkage ratio of natural cities in the Yellow River affected area.

| | 100 km$^2$ | | 10 km$^2$ | |
| --- | --- | --- | --- | --- |
| | $\geq$100 km$^2$ | <100 km$^2$ | $\geq$10 km$^2$ | <10 km$^2$ |
| Mild shrinkage | 1.06% | 98.94% | 29.08% | 70.92% |
| Severe shrinkage | 2.48% | 97.52% | 32.30% | 67.70% |

### 3.2.3. Overall Spatial Pattern of Growth

In recent years, with the acceleration of China's urbanization process, many cities have developed in a disorderly mode, which has also affected most cities in the Yellow River affected area. As shown in Figure 6, the expansion cities are mainly distributed in the middle and lower reaches of the Yellow River affected area where there are more cities. Except for a small number of shrinking and stable cities, most of the cities in the lower reaches are expansion cities. In addition, the administrative cities of Lanzhou and Yinchuan in the upper reaches of the Yellow River affected area also have an obvious expansion trends.

The expansion rates of natural cities in the Yellow River affected area reached up to 69.17% between 2013 and 2018. Among the expanding natural cities, the number of cities with mild expansion accounts for 17.64%, among which those with a natural area of more than 100 km$^2$ account for 1.83%, those with a natural area of less than 100 km$^2$ account for 98.94%, those with a mild expansion of more than 10 km$^2$ account for 29.32%, and those with a natural area of less than 10 km$^2$ account for 70.68%. The number of cities with significant expansion accounts for 82.36%, of which 1.35% are cities with natural urban area of more than 100 km$^2$, and 98.65% are cities with natural urban area less than 100 km$^2$. In addition, 22.99% of the cities with an area of more than 10 km$^2$ and 77.01% of the cities with less than 10 km$^2$ (Table 4). Urban expansion is an inevitable stage in the process of urbanization, which is not only the result of market selection, but also the result of a government's active spatial intervention. Since the lower reaches of the Yellow River affected area are mostly plains, and some cities are distributed along the Yellow Sea and the Bohai Sea, with convenient traffic conditions, more ports and developed local, regional and national transportation hubs. In addition, with the support of national economic development strategy, coastal cities radiate to drive the development of surrounding cities, which makes the cities in the lower reaches of the Yellow River affected area develop rapidly and expand significantly. The administrative cities of Lanzhou and Yinchuan, in

the upper reaches of the Yellow River affected area, are the capitals of Gansu Province and Ningxia Hui Autonomous Region, respectively. Resources in all aspects are relatively concentrated, which has led to rapid economic development in the two regions and rapid urban expansion. In general, the expansion of natural cities in the Yellow River affected area is that the number of expanding cities in the lower reaches is much greater than in the middle and upper reaches.

**Table 4.** Expansion ratio of natural cities in the Yellow River affected area.

| | 100 km$^2$ | | 10 km$^2$ | |
|---|---|---|---|---|
| | $\geq$100 km$^2$ | <100 km$^2$ | $\geq$10 km$^2$ | <10 km$^2$ |
| Mild expansion | 1.83% | 98.94% | 29.32% | 70.68% |
| Significant expansion | 1.35% | 98.65% | 22.99% | 77.01% |

## 4. Discussion

Different from the previous studies on the definition of shrinking and expanding cities at the administrative city level based on the traditional statistical data, we define the natural city as the benchmark, which is more in line with the spatial physical structure of a city, thus providing a broad and accurate method to describe and understand the latest distribution characteristics of shrinking and expanding cities. The redefined natural city transcends the limitations of the independent development of multiple urbanized areas within an administrative city, and helps explain the changes in each natural city. This study uses quantitative methods to determine the shrinking and expanding areas of a city. The results of the study can effectively track changing characteristics and differences of human activity from place to place in a city, thereby guiding the renewal and restoration of a city and the structural adjustment within a big city.

In addition, there are still some deficiencies in this work that need to be improved. Due to the change of Light Emitting Diode (LED) street lamps, the night light spectrum changes [94,95]. The wavelength of the light emitted by the white LED lights is typically less than 500nm (blue), and the transition of street lamp from (Orange) High Pressure Sodium (HPS) lamp to LED lamp with constant surface brightness could lead to certain decreases in radiation observed by the night light sensors. For this reason, 30% of the nighttime light brightness decline might be due to the transition from HPS to LED lights, rather than the real decline of nighttime lighting [96], and therefore the urban changes identified cannot be as accurate as conceptually expected. Compared with the results of the same period of LandScan data, there are some differences in the distribution of shrinking and expanding cities, the number of changed cities identified by LandScan is significantly less than that of night light data. Although the night light data proves their accuracy in identifying urban shrinkage and expansion, the method of calculating the slope of the change in the brightness of each pixel at night still contains certain uncertainty. The slope change of night light brightness value is used to identify the shrinking and expanding cities, which could underestimate the change of the weak night light area. The reason is that, for the areas with weak night light, the calculated brightness change slope can be also small, which means that most of these areas are stagnant and slightly shrinking. In addition, due to the rapid development of urban economy in the strong nighttime lighting area, the calculated nighttime light brightness change slope is greater than 1, and these results are all taken as 1 for calculation in statistical calculation. As data products are updated and improved, we will extend the research cycle to more accurate identification and mapping of shrinking and expanding cities.

The shrinkage or expansion of a city is not determined by a single factor but the effects of the superposition of various external factors. The reasons for the shrinkage of different types of cities are different. The shrinkage of small and medium-sized cities in coastal developed areas could be caused by the low-end manufacturing industries as their industrial base; for example, their industrial structure has gradually shifted from

the manufacturing industry to information, technology, and tertiary industries, while the low-end manufacturing industry has been gradually transferred to Southeast Asia, where cheaper labor can be available, resulting in the shrinkage of small and medium-sized coastal manufacturing cities [97]. Because coal mine resources are gradually exhausted, there is already a general trend of the gradual decline of urban coal mining industry in resource dependent cities. Small and medium-sized industrial cities based on coal mining industry are unable to provide jobs, resulting in a large number of population loss [98,99]. Absorbing the population of surrounding villages and towns as a supplement, large cities have convenient public service facilities and cause the population of surrounding small cities to flock into large cities, and the shrinkage of small cities occurs. The population of small and medium-sized cities in land has not been greatly reduced, because it is still in the initial stage of contraction. However, the increase in population of large cities has caused significant urban sprawl [100–102]. Therefore, the factors of urban contraction and expansion need to be further explored in future research.

## 5. Conclusions

By redefining the natural city and using the night light data to identify shrinking and expanding city, we analyzed and mapped the spatial distribution of the shrinking and expanding cities. We use a changing rate method to identify shrinking and expanding cities, and based on POI data and road network data, this method redefines 3130 natural cities in the Yellow River affected area. Compared with the administrative cities used in previous studies, our natural city conception and method successfully identify changing cities, which conforms to Zipf's law very well. We have identified 604 shrinking cities based on night light data, of which 322 are severely shrinking and 282 are slightly shrinking. The severely shrinking cities will be the focus of future urban planning, as the severely shrinking cities will leave a large amount of vacant land. Of course, the planning of expanding cities is also very important. The shrinking cities are mainly distributed in the upper reaches of the Yellow River affected area and small cities around the central cities of several major provincial capitals. 2165 cities have been identified for expansion, of which 1783 have significantly expanding and 382 have slightly expanding. The expanding cities are mainly located in the middle and lower reaches of the Yellow River affected area and Lanzhou and Yinchuan in the upper reaches. There are 361 cities that have been identified as basically stable, and those cities are growing slowly. The identification of shrinking and expanding cities and the analysis and summary of their spatial distribution provide a feasible reference for planning and management departments in formulating an urban development planning outline.

For further applications of NTL data, the identification of weak light area shrinkage and strong light area expansion needs to be improved in future, because the method used in this paper could underestimate the shrinkage of weak light area and the expansion of strong light area. At the same time, this result shows us that it is necessary to conduct field investigation to understand the causes of the generation and changes of urban nighttime lighting. In particular, the cause of the NTL outliers can be checked for areas with outliers, rather than smoothing or removing outliers. In addition, this paper takes the cities in the affected area of the Yellow River as an example to identify the shrinking and expanding cities and analyze their spatial distribution. In future work, we will further expand the study to the interior of a city and constantly explore the law of urban change.

**Author Contributions:** This research was carried out under the cooperation of all authors. Haoming Xia provided the writing ideas for the research, Wenhui Niu completed data collection, analysis and wrote the paper, and Haoming Xia, Qingmin Meng, Ruimeng Wang, Li Pan, Yaochen Qin, Rumeng Li, Xiaoyang Zhao, Xiqing Bian and Wei Zhao all contributed to the discussion and revision of the paper. All authors have read and agreed to the published version of the manuscript.

**Funding:** This research was funded by Henan Provincial Department of Science and Technology Research Project (212102310019) and the major project of Collaborative Innovation Center on Yellow

River Civilization jointly built by Henan Province and Ministry of Education (2020M19). We are grateful to all contractors, image providers.

**Acknowledgments:** We thank the editors and the anonymous reviewers for their valuable comments and suggestions.

**Conflicts of Interest:** The authors declare no conflict of interest.

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
