# Peer review of "Research on Large-Scale Urban Shrinkage and Expansion in the Yellow River Affected Area Using Night Light Data"

_ijgi, doi:10.3390/ijgi10010005_

Round 1

Reviewer 1 Report

Overall, the manuscript is mainly well written but extensive English editing is still required. The use of past/present tense is not consistent.

Some particular comments:

Line 39: are 20-year-old papers still state of the art?

Line 39: “schlumbfende St è 40  DTE" - While this may sound like a perfectly good German term, I can confidently say that it is not. The correct spelling is “schrumpfende Stadt“. Please note the r (not l) and p (not b) in “schrumpfende“. I assume the jumbled second word originally read “Städte“ (cities, plural form); best to leave the singular form “Stadt“. I’d like to see a reference for the origin of the term.

Line 44: Since the reform and opening up - some more detail for an international readership would be helpful.

Line 46: please be more specific. “Some problems“ sounds very vague.

Line 90: socioeconomic: please be consistent with the spelling; avoid mixing (socio-economic, social economic, socioeconomic).

Line 104: why do natural cities have be redefined? The intent of the authors is not clear. The concept of a natural city needs to be further explained at this point.

Line 105-108: why is it “easy to express the development status of an individual city“ - please specify. Also, how can uneven development be described? This paragraph needs to be revised grammatically and more information is required for the reader to understand.

Line 203-208: the five steps need further explanation, especially the use of the thresholds. Were other thresholds also tested, and if so, with what results? Also, lines 205/206 are misleading because the POI density of >50 points/km2 was selected, not screened out. What difference did the road data make (see also comment below)? What kind of “correction” did that achieve?

Line 119: more information for an international readership is required.

Figure 1 b and c: in this representation of the night time imagery, there is no obvious difference between the two years apart from the maximum pixel value in the legend. Is there a more recognizable difference with a different gradient of the grey values? Otherwise some differences should be explained in the caption.

Line 129/130: are these websites available in English too? Otherwise an international readership will not be able to access these resources to recreate the results. What is represented by the POIs? Any specific categories? And what information is in the road network data - all roads including narrow footpaths or specific types of roads only?

Line 140: this is a strong statement that needs supporting verification.

Line 142: again (see comment above), what do the POIs and road data represent? The total amount of “road data“ does not mean anything if a road consists of several smaller (or larger) segments.

Line 149-153: needs to be rewritten - incomprehensible and grammatically incorrect.

Line 192: more evidence for this - “more and more scholars“ requires more than one citation

Line 239/240: why? An explanation for the focus on geographical selection is necessary. Just because there’s more is too arbitrary.

Figure 6: color scheme needs to be changed from qualitative to diverging for a meaningful visual display. Otherwise this is hard to interpret.

Line 266-280: a table would support the readability of this paragraph, rather than too much text.

Line 285: why are NTL data necessarily more accurate? How can this be objectively verified? NTL data measure light emissions, the results of this study are only an interpretation of these data. The authors need to support their statement with stronger evidence rather than “more can be identified”.

Line 293: NTL imagery is also a proxy of economic performance - although this was mentioned earlier, it needs to be addressed again at this point to avoid what seems like a contradiction at this point.

Line 364-370: is this the case in the study area? The authors quote general, global studies rather than give evidence how this affects the study area.

Line 391/392: why only shrinking cities? Isn’t urban planning also necessary in expanding cities? Give reasons here.

Line 423: scalling - change to scaling

Line 424: nancial - incorrect reference; please change to financial

Line 439: incorrect referencing. Ostfildern is a place; please update quotation with correct data fields

Line 441: Zietschrift - change to Zeitschrift

Reviewer 2 Report

I liked the written paper. Its content and research are modern, interesting and useful. Used methodology is a good way to get information about a very important data looking on many aspects of life, demographic and environmental. I have only two minor remarks. One is that in key words I believe Night shouldn't be written with capital letter. The other is in lines 126-130 there are some spaces missing before the brackets.

Reviewer 3 Report

This paper investigates a very important issue of urban growth, stagnation, shrinkage dynamics that are typically experienced in fast-developing countries worldwide. What's novel of the presented research is that it identified both shrinkage and growth (or sprawl) simultaneously, while classifying the degree of shrinkage into categories by method (NTL & pop data). One of the important findings is that small-scale shrinkage occurs actively despite the significant economic growth of the region. The paper is overall clearly written with a robust methodology. I have two minor suggestions as below:

(1) A discussion on possible underlying factors affecting small cities to struggle and larger cities to grow even larger would be of additional value, particularly in Chinese context. Consider providing an social/economic/historical explanation such as Straw Effect where transportation infrastructure investment that improves access to amenities and key public services in big cities result in smaller cities losing their population and services. 

(2) A lit review is missing. I understand NTL is growing in popularity in identifying urban activities, but there has been research that uses other methods to understand urban patterns. Provide a brief discussion why NTL is chosen and superior to prior methods in the field including:

-Kuffer, M., Pfeffer, K., Sliuzas, R. and Baud, I., 2016. Extraction of slum areas from VHR imagery using GLCM variance. IEEE Journal of selected topics in applied earth observations and remote sensing, 9(5), pp.1830-1840.
-Park, Y. and Guldmann, J.M., 2020. Measuring continuous landscape patterns with Gray-Level Co-Occurrence Matrix (GLCM) indices: An alternative to patch metrics?. Ecological Indicators, 109, p.105802.
-Liu, Y. and Jiang, Y., 2020. Urban growth sustainability of Islamabad, Pakistan, over the last 3 decades: a perspective based on object-based backdating change detection. GeoJournal, pp.1-21.

Round 2

Reviewer 1 Report

The paper has been improved since the first version, however there are still some issues that need to be addressed.

The manuscript still needs to be checked for typographic errors such as missing or extra spaces, particularly before and after punctuation or special characters.

Please do not copy and paste the same sentence at the beginning of each answer.

Lines 44-45: I still feel that „some environmental and ecological problems“ is very random; these problems need to be further explained. If they have „complex impacts“ they need to be addressed in more detail.

Line 105: where does the limit of 2 km2 come from? Quote from literature or reason for this limit needs to be added.

Line 109: better to use „administrative unit“ rather than „administrative city“

Line 124: this sentence is hard to read; suggestion: „… cities that rely on these resources to deteriorate economically…“

Line 206-212: This is still not adequately described; the answers given in the response to the authors need to be included here. 

Line 133: same comment as in the previous review; the POIs and road network data need further explanation (no full list required, examples are sufficient).

Point 15 in the previous review was not addressed at all although the authors claimed it has been rewritten.

Figure 6: I’m still not entirely happy with the representation in the map; a diverging color scheme is not necessarily continuous. Having distinct categories is ok, however it needs explanation how the original mild/severe shrinkage/expansion were derived and the color scheme needs to reflect that. 

Response 23 from the previous review needs to be incorporated in the manuscript. Otherwise it is still not clear enough why the authors focus on shrinking cities.

Author Response

Review Report (Round 2)

We greatly appreciate these valuable comments from the editor and reviewers, which help us to improve our study. Following your comments, we made a careful revision of this manuscript. In the following text, our responses to the reviewers’ comments are in red. The revisions are highlighted using the “Track Changes” function in Microsoft Word.

Minor comments:

Point 1: The manuscript still needs to be checked for typographic errors such as missing or extra spaces, particularly before and after punctuation or special characters.

Response 1: We are very sorry for this careless mistake. We have carefully examined and revised the manuscript for typographical errors.

Point 2: Please do not copy and paste the same sentence at the beginning of each answer.

Response 2: Thank you very much for your suggestion, and we will accept it without using the same sentence at the beginning of each answer.

Point 3: Lines 44-45: I still feel that “some environmental and ecological problems” is very random; these problems need to be further explained. If they have “complex impacts” they need to be addressed in more detail.

Response 3: We have further explained this problem in the manuscript, “urban expansion has caused a series of ecological and environmental problems, such as the impact on the primary net productivity of urban vegetation, urban water evolution, and urban heat island effect”.

Point 4: Line 105: where does the limit of 2 km2 come from? Quote from literature or reason for this limit needs to be added.

Response 4: Thank you very much for your suggestions to help us improve the manuscript. The limitation of 2km2 we used in the manuscript is based on the article “Are all cities with similar urban form or not? Redefining cities with ubiquitous points of interest and evaluating them with indicators at city and block levels in China”, and we have added references in the manuscript.

Point 5: Line 109: better to use “administrative unit” rather than “administrative city”.

Response 5: We have accepted your suggestion to replace the word " administrative city " with " administrative unit ".

Point 6: Line 124: this sentence is hard to read; suggestion: “… cities that rely on these resources to deteriorate economically…”.

Response 6: Thank you very much for your comments on our manuscript. We have revised this sentence according to your comments.

Point 7: Line 206-212: This is still not adequately described; the answers given in the response to the authors need to be included here.

Response 7: We have supplemented the answers in the manuscript with a detailed description of the process of defining natural cities.

Point 8: Line 133: same comment as in the previous review; the POIs and road network data need further explanation (no full list required, examples are sufficient).

Response 8: We have explained POI and road network data in the form of table in the manuscript, and the data list is as follows.

POI and road network data type example table

Name

Point of interest

Road network

Catering services

Highway

Public facilities

National road

Shopping services

Provincial road

Road ancillary facilities

County road

Companies and enterprises

Township road

Toponym and address information

Pedestrian road

Point 9: Point 15 in the previous review was not addressed at all although the authors claimed it has been rewritten.

Response 9: We are very sorry for this mistake. We have made some modifications to the questions raised and rewritten the paragraph in the manuscript.

Point 10: Figure 6: I’m still not entirely happy with the representation in the map; a diverging color scheme is not necessarily continuous. Having distinct categories is ok, however it needs explanation how the original mild/severe shrinkage/expansion were derived and the color scheme needs to reflect that.

Response 10: Thank you very much for your insightful comments. We use different color schemes to represent different categories in the manuscript. According to formula (1), we calculate the change rate of cities. The change of cities can be explained as follows: the shrinkage rate of cities with severe shrinkage is - 100% – - 30%, that of slight shrinkage is - 30% – - 10%, that of slight expansion is 10% – 30%, and that of significant expansion is 30% – 100%.

                        (1)

Point 11: Response 23 from the previous review needs to be incorporated in the manuscript. Otherwise it is still not clear enough why the authors focus on shrinking cities.

Response 11: Thank you very much for your proposal. We have incorporated the answers in response 23 of the previous comment into the manuscript.
